# Electrically Conductive Nanocomposites Composed of Styrene–Acrylonitrile Copolymer and rGO via Free-Radical Polymerization

**DOI:** 10.3390/polym12061221

**Published:** 2020-05-27

**Authors:** Eun Bin Ko, Dong-Eun Lee, Keun-Byoung Yoon

**Affiliations:** 1Department of Polymer Science and Engineering, Kyungpook National University, Daegu 41566, Korea; ebko@naver.com; 2School of Architecture & Civil Engineering, Kyungpook National University, Daegu 41566, Korea

**Keywords:** polymerizable reduced graphene oxide, in situ polymerization, electrical conductivity, dispersion of 2D nanosheets

## Abstract

The polymerizable reduced graphene oxide (mRGO) grafted styrene–acrylonitrile copolymer composites were prepared via free radical polymerization. The graphene oxide (GO) and reduced graphene oxide (rGO) was reacted with 3-(tri-methoxysilyl)propylmethacrylate (MPS) and used as monomer to graft styrene and acrylonitrile on its surface. The successful modification and reduction of GO was confirmed using Fourier transform infrared spectroscopy (FT-IR), thermogravimetric analyzer (TGA), Raman and X-ray diffraction (XRD). The mRGO was prepared using chemical and solvothermal reduction methods. The effect of the reduction method on the composite properties and nanosheet distribution in the polymer matrix was studied. The thermal stability, electrical conductivity and morphology of nanocomposites were studied. The electrical conductivity of the obtained nanocomposite was very high at 0.7 S/m. This facile free radical polymerization provides a convenient route to achieve excellent dispersion and electrically conductive polymers.

## 1. Introduction

Graphene oxide (GO) is one of the most promising fillers for nanocomposites, due to its unique electrical, mechanical, optical and thermal properties [1,2,3,4,5]. GO sheets have been thoroughly investigated due to their good dispersity and possible post-functionalization. Because of they are easily reduced, they provide tunability of electrical conductivity [6,7].

The reduction of GO offers the potential to produce few-layered graphene sheets, i.e., reduced GO (rGO), in high amounts. It also offers the possibility to process advanced graphene-based materials, consisting of various oxygen containing functional groups and for any application concerning conductive polymer composites [8,9,10]. There are several reduction processes used to prepare rGO, including chemical reduction, solvothermal reduction, electrochemical reduction, thermal reduction and the use of surfactants or stabilizers. Among these, chemical reduction (especially with hydrazine) and solvothermal reduction are among the most effective and convenient approaches for preparing processable colloidal suspensions required for a broad field of applications [11,12,13].

Because rGO has high thermal and electrical conductivity and rapid transfer is possible between the effectively contacted rGO lamellae, the conductivity of nanocomposites can be significantly improved [14,15]. Therefore, the relative content of rGO within nanocomposites must be increased to form a continuous phase. Unfortunately, aggregation of rGO can easily occur with increasing rGO content. In earlier studies, solution blending, melt mixing and in situ polymerization have been used to produce these polymer/rGO composites [16,17,18,19]. In situ polymerization is commonly performed by mixing the filler in the presence of monomers, followed by subsequent polymerization. Upon this method, it is possible to obtain uniformly dispersed polymer/rGO nanocomposites.

In recent years, the grafting of GO sheets with polymer was studied for well-dispersed GO in polymer matrix. Yao et al. [20] grafted polyacrylonitrile using free radical polymerization on gamma irradiation. Yang et al. [21] prepared polystyrene/rGO nanocomposites with radical polymerization, the obtained nanocomposites were enhanced thermal stability on a uniform dispersion of rGO with polymer matrix. Voylov et al. [22] reported that poly(sodium 4-vinylbenzenesulfonate) obtained via RAFT had a controlled molecular weight with a very narrow polydispersity. Sánchez et al. [23] prepared rGO grafted with relatively short chains of poly(n-butyl methacrylate) for electrical field grading materials on high-voltage direct-current applications.

Such in situ polymerization and grafting polymerization methods provides the good dispersion and compatibility of rGO in polymer matrix, and thus excellent thermal, mechanical and electrical properties of the nanocomposites. Researches of polymeric nanocomposite using various monomers were reported, however few studies gave been done on GO-grafted radical copolymerization of styrene and acrylonitrile. Styrene–acrylonitrile copolymer (SAN) is especially tough and resistant to chemicals, it used a broad range of industries including the food, construction and electrical engineering.

In this work, we demonstrate an alternating approach for the growth of polymers from the surface of GO by covalently attaching polymerizable acrylate monomer and reducing them using reducing agents and solvothermal methods. GO was functionalized with 3-(trimethoxysilyl)propyl methacrylate (MPS), and subsequently, this polymerizable mGO was reduced using chemical and solvothermal reduction to prepare the modified reduced graphene oxides (mRGOs). The grafting of graphene with styrene–acrylonitrile copolymers was performed via conventional radical polymerization in the presence of mRGOs. The effects of each reduction method on the electrical conductivity, thermal stability and mechanical properties of the nanocomposites were examined.

## 2. Experimental

### 2.1. Materials

Expanded graphite (Timcal Graphite & Carbon, Paris, France, <100 μm, 99.9%), sulfuric acid (H_2_SO_4_, Duksan Chemical Co., Ansan, Korea, H_2_SO_4_, ≥95%), hydrochloric acid (HCl, Duksan Chemical Co., Ansan, Korea, HCl, 37%), hydrogen peroxide (H_2_O_2_, Duksan Chemical Co., Ansan, Korea, H_2_O_2_, 28%), sodium nitrate (Daejung Co., Seoul, Korea), potassium permanganate (Sigma-Aldrich, St. Louis, MO, USA, >99.0%), hydrazine monohydrate (Junsei, Tokyo, Japan, >98.0%), 3-(trimethoxysilyl)propyl methacrylate (MPS, TCI, Tokyo, Japan, >98.0%), dimethyl formamide (DMF, Duksan Chemical Co., Ansan, Korea, 99.9%) and 1-methyl-2-pyrrolidinone (NMP, Daejung Co., Seoul, Korea, 99.5%) were used without further purification. Styrene (Junsei, Tokyo, Japan, >99.5%) and acrylonitrile (Junsei, Tokyo, Japan, >99.0%) were purified by stirring with calcium hydride and distilling under reduced pressure. Azobisisobutyronitrile (AIBN, Sigma-Aldrich, St. Louis, MO, USA, 98%) was recrystallized from ethanol.

### 2.2. Synthesis of Graphene Oxide, Modified GO and Reduced mGO

GO was prepared from expanded graphite following Hummers oxidation method [24,25]. The polymerizable modified GO (mGO) was prepared as follows; GO (1 g) was dispersed in DMF (200 mL) via ultrasonication for 12 h and subsequently, MPS (20 mmol) in DMF was added to this dispersion, in a dropwise manner for 0.5 h. The mixture was stirred for 24 h at 60 °C to allow the formation of covalent linkage between the hydroxyl groups on GO and the silyl group. Subsequently, methanol (100 mL) was added to remove any unreacted MPS. Therefore, it was washed sequentially with methanol and DMF.

The mRGOs are produced via two different methods of reduction; chemical (mRGO1) and solvothermal (mRGO2) reduction methods, which are detailed as follows [26,27]. The chemical reduction was carried out by adding hydrazine monohydrate (1 mL) to the mGO suspension, which was stirred at 70 °C for 24 h. The color of suspension was observed to change from brown to black, indicating that the reduction of mGO had occurred. The reaction mixture was filtered and washed repeatedly with DMF to remove any excess, unbound hydrazine. A suspension of mRGO1 was finally obtained via ultrasonication in DMF (200 mL).

For the solvothermal reduction method, a suspension of GO (1 g) in DMF (200 mL) was heated to 180 °C and refluxed for 24 h. It was subsequently filtered and washed with DMF several times. The rGO (1 g) was re-dispersed in DMF (200 mL) via ultrasonication for 12 h and MPS (20 mmol) was subsequently added dropwise 0.5 h. The resulting mixture was stirred for 24 h at 60 °C to facilitate the occurrence of silylation. Subsequently, 100 mL of methanol was added to remove any residual silane molecules. The product was washed sequentially with methanol and DMF. The suspension of mRGO2 was finally obtained via ultrasonication in DMF (200 mL). The reaction procedure was illustrated in Scheme 1.

### 2.3. In situ Copolymerization of Styrene and Acrylonitrile

Radical polymerization was conducted in a 500 mL glass reactor equipped with a magnetic stir bar. The reactor was back-filled three times with nitrogen and charged with styrene, acrylonitrile, mGO, mRGO1 and mRGO2. The mixture was heated to 80 °C and AIBN was added and subsequently, it was stirred under nitrogen for 24 h.

After the polymerization was complete, the mixture was poured into an excess of methanol to precipitate the polymer. The grayish powder was filtered, washed with methanol and dried overnight at 60 °C under vacuum.

The grafting ratio was determined by Soxhlet extraction. The molecular weight, thermal stability and electrical conductivity of the obtained copolymer were used without extraction.

### 2.4. Characterizations

The chemical structures of GO, mGO and mRGOs were examined using Fourier transform infrared (FT-IR) spectroscopy (Jasco 4100, Tokyo, Japan) and Raman spectroscopy (Nicolet Almega XR, Thermo Scientific, Waltham, MA, USA).

The X-ray diffraction (XRD) patterns were obtained using Philips X-Pert PRO MRD diffractometer (Malvern Panalytical Ltd., Malvern, UK) equipped with Cu-Kα radiation.

The thermal stability was tested at 20 °C/min from 30 to 800 °C under nitrogen atmosphere using a Setaram Labsys evo thermogravimetric analyzer (TGA, Setaram Instrumentation, Paris, France).

The electrical conductivity was measured by the four-point probe method using a Keithley 2400 semiconducting characterization system (Kiethiley, Cleveland, OH, U.S.) at 25 °C. The volume conductivities of the polymer and copolymers with resistance higher than 10^7^ Ω were measured using CHI 660E electrochemistry workstation (CH Instruments, Inc., Beijing, China).

The molecular weight of the obtained polymer and copolymers were determined by using an Ubbelohde viscometer (Sigma-Aldrich Co., St. Louis, MO, USA) in tetrahydrofurane at 24 °C.

The morphologies of the GO, mRGOs and the fractured surfaces of nanocomposites were characterized by a field-emission scanning electron microscope (FE-SEM, JSM-6380LV, Joel, Tokyo, Japan).

## 3. Results and Discussion

### 3.1. Characterization of mGO and mRGOs

GO was synthesized from graphite using Hummers’ method. MPS was used to modify GO to yield MPS-modified GO (mGO), which was subsequently reduced via chemical reduction with hydrazine (mRGO1). GO was also reduced via solvothermal reduction using DMF followed by silylation with MPS (mRGO2).

As shown in Figure 1, FT-IR spectroscopy was used to confirm the structures of GO, mGO and mRGOs and the formation of covalent bonds between the components. The GO exhibited peaks at 1040 cm^−1^, 1610 cm^−1^, 1720 cm^−1^ and 3300 cm^−1^ corresponding to the C–O, C=C, C=O and –OH groups, respectively, confirming that it contains a variety of oxygen-containing functional groups. For MPS modified materials, methylene C–H stretching peak appeared at 2920 cm^−1^ and 2850 cm^−1^, methyl C–H deformation peak at 1465 cm^−1^ and Si–O–C stretching peak at 1130 cm^−1^, also confirming that MPS was successfully introduced on the surface of GO. After the modification of GO with MPS and its reduction, the –OH stretching at 3300 cm^−1^ disappeared and the characteristic bands at 1300 cm^−1^ (C–Si stretching) and 1030 cm^-1^ (Si–O–C stretching) of the mGO appeared [28].

Raman spectroscopy and XRD were performed to determine the ordered and disordered carbonaceous materials and the results are shown in Appendix A, respectively. The D/G band intensity ratio (*I_D_/I_G_*) of mRGO1 and mRGO2 were 1.37 and 1.13, respectively, which were higher than that of GO (0.98). It indicated that the number of aromatic domains increased after the reduction step (in Appendix A) [29]. After the modification of GO with MPS, the diffraction angle in XRD patterns (in Appendix A) decreased from 11.3° to 10.4°, corresponding to an increase in the interlayer spacing from 0.78 to 0.85 nm. However, these peaks disappeared after the reduction of mGO. These phenomena indicate that MPS is bonded on the surface of GO successfully and is capable of enlarging the interlayer distance between GO sheets. For mRGO1 and mRGO2, a new broad peak can be observed at 22°–24°, which are close to that of pristine graphene nanosheet (25°) [30]. The results of Raman spectroscopy and XRD as shown in observation of FT-IR, MPS was successfully introduced on the surface of GO and the chemical and solvothermal method effectively reduced GO.

To observe the morphologies and layers of the obtained GO, rGO, mGO and mRGO, SEM was performed, and the images shown in Appendix A.

Optical observation is a direct way to see the dispersion of mRGOs. Figure 2 shows the dispersion states for mRGOs just sonication, after 18 h, 48 h and 144 h, respectively. All mRGOs showed good dispersion in DMF after just sonication. However, after 18 h, GO and mGO started to precipitate and after 144 h, they showed significant precipitation in DMF. The dispersion of mRGO1 and mRGO2 in the DMF did not precipitate even after 144 h from sonication, which attributed to the exfoliation of mRGO1 and mRGO2 layers.

The thermal stability of GO, mGO and mRGOs was measured using TGA in a nitrogen atmosphere as shown in Figure 3. The GO had lower thermal stability than mGO and mRGOs and exhibited three distinct weight loss steps. The first step is the weight loss until 150 °C, and was caused by the residual evaporation of water (approximately 10%); the second step was from 150 to 300 °C (approximately 34%) and was attributed to the decomposition of labile oxygen-containing groups such as carboxylic, anhydride or lactone groups forming CO, CO_2_ and steam; the third step above 300 °C (approximately 16%) was related to the pyrolysis of the carbon skeleton of GO [31]. Functionalizing the surface of GO with MPS and subsequent reduction processes caused the weight loss of approximately 12 wt % for mGO, 12 wt % for mRGO1 and 5 wt % for mRGO2 over 150–300 °C, mainly attributed to the removal of oxygen-containing functional groups. Further, mRGOs also exhibited higher char yields, 70 wt % for mRGO1 and 80 wt % for mRGO2. The solvothermally reduced GO (mRGO2) showed similar characteristics, but with lower amount of weight loss, compared to that of mRGO1. This could be explained by a smaller amount of oxygen functional groups in the surface.

The electrical conductivity of mGO and mRGOs was measured using a four-point probe method; the results are shown in Figure 4. The reduction process increased the conductivity up to 9.8 × 10^3^ S/m owing to the restored π-conjugated system in the mRGO film. Several factors are influenced by the electrical conductivity of graphene derivatives such as the amounts of defects, reduction atmosphere, film thickness and residual oxidation [32,33,34]. The electrical conductivities of mRGO1 and mGRO2 were 7.6 × 10^3^ S/m and 9.8 × 10^3^ S/m, respectively. The reduction methods did not significantly affect the conductivity, however, mRGO2 has a slightly higher electrical conductivity than mRGO1 because mRGO1 contains more oxygen functional groups in surface as observed in the TGA analysis.

### 3.2. Preparation of SAN/rGO Nanocomposites via in situ Radical Polymerization

Table 1 presents the results of the copolymerization of styrene and acrylonitrile in the absence or presence of mRGO initiated by AIBN. The grafted copolymer was named SAN–mGO, SAN–mRGO1 and SAN–mRGO2 for the obtained copolymer using mGO, mRGO1 and mRGO2, respectively.

With the introduction of mRGOs, the conversion of the monomer decreased slightly and the mRGOs contents was varied between 3.3 wt % to 16.0 wt %. The concentration of mRGO and molecular weight of the copolymers did not affect the monomer conversion. The molecular weight was in the range of 5.0 × 10^4^ to 8.0 × 10^4^ g/mol.

The compositions of styrene and acrylonitrile were calculated using a calibration curve, obtained via FT-IR spectroscopy. The absorbance of the C=N stretching vibrational peak (2250 cm^−1^) and the aromatic C=C stretching vibrational peak (1450 cm^−1^) are characteristic of acrylonitrile and styrene, respectively. The calibration curve was obtained from the intensity ratio (*I_2250_/I_1450_*) of the mixture of polyacrylonitrile and polystyrene. The comonomer composition of the copolymers was calculated using the obtained calibration curve. (see Appendix A) The mole fraction of styrene in the copolymer was in the range of 68–74 mol%.

In order to measure the grafting efficiency, the nanocomposites obtained by free-radical polymerization was extracted as methyl ethyl ketone (MEK) using a Soxhlet apparatus [35]. Most of the copolymer that was grafted on mGO and mRGO was extracted only 5–6 wt %, while more than 98 wt % of copolymers without mGO and mRGO was extracted. Although there is a limit to calculating grafting efficiency with simple extraction, the obtained grafted copolymer was used to measure thermal stability and electrical conductivity because the extraction amount was only about 5 wt %.

The effect of the mRGOs on the thermal stability of the obtained styrene–acrylonitrile copolymers was examined using TGA in a nitrogen atmosphere.

TGA thermograms of SAN, SAN–mGO, SAN–mRGO1 and SAN–mRGO2 with different m(R)GOs contents are shown in Figure 5. The degradation temperature at 5% and 10% weight loss (*T*_d5%_ and *T*_d10%_) and the char yields are summarized in Table 2.

The weight loss curves exhibited a one-step degradation process and shifted slightly to higher temperatures with the introduction of mGO and mRGOs. The degradation temperatures of grafted nanocomposites (*T*_d5%_) are higher than that of neat copolymer. For 1.8, 3.0 wt % and 4.7 wt % mRGO2 contained SAN–mRGO2 nanocomposites, the *T*_d5%_ of composites was 23.5, 24.5 and 28.8 °C higher than that of SAN copolymer. In case of SAN–mRGO1, the *T*_d5%_ of composites increased by more than 20 °C with introduction of mRGO1. A distinct increase in thermal stability was due to mRGO being well-dispersed in the copolymer matrix. The improvement in thermal stability can be attributed to the delays the escape of volatile degradation products and thus slows down the initial degradation step [36]. The char yield of grafted nanocomposites was slightly higher than the amount of mRGO. The case of SAN-mGRO2 containing 16 wt % of mRGO2 was 25 wt %. The thermal stability of the obtained nanocomposites was almost same values even with mRGO1, which the loading amount of the MPS is twice as much as mGRO2. These results are not due to MPS modification, but to the high heat capacity and thermal conductivity of mRGO nanosheets, which can form the heat-resistant layers that act as mass transfer barriers [37].

The SAN–mRGO2 nanocomposites have excellent thermal stability, which can be attributed to good interfacial interactions between mRGO2 and SAN. From the SEM image of SAN–mRGO2, the well-dispersed status of mRGO2 in copolymer matrix with little aggregation, suggesting good adhesion between the fillers and matrix (Figure 7). Due to the interfacial interaction between mRGO2 and copolymer, the char was easy to form on the surface of mRGO sheets, which resulted in the high char yield of nanocomposites [38].

Superior electrical conductivity is the most important feature of graphene. It is well known that good dispersion of fillers is important for polymer nanocomposites to form a conducting network within the polymer matrix [39]. Polymer/graphene nanocomposites generally exhibit a nonlinear increase in electrical conductivity as a function of the filler concentration. At a certain filler loading fraction, which is known as the percolation threshold, the fillers form a network leading to a sudden rise in the electrical conductivity of the composites [40,41]. Patole et al. [42] prepared polystyrene/graphene films, containing 3.0 wt % graphene with a conductivity of 4.5 × 10^–5^ S/m. Tripathi et al. [43] also observed that the conductivity of in situ polymerized polymethylmethacrylate/rGO nanocomposites at 2.0 wt % loading of rGO was 9.9 × 10^–5^ S/cm.

Figure 6 shows the electrical conductivity of SAN–mGO, SAN–mRGO1 and SAN–mRGO2 as a function of mRGO concentrations. The electrical conductivity of SAN–mRGO1 and SAN–mRGO2 composites increased significantly, demonstrating a sharp transition from an electrical insulator to conductor with low percolation thresholds of 1.9 wt % for SAN–mRGO1 and 1.8 wt % for SAN–mRGO2. The electrical conductivity of SAN–mRGO1 significantly increased by five orders of magnitude with the increases in the concentration of mRGO1 from 1.9 wt % to 3.3 wt %, with a value of 7.5 × 10^–3^ S/m. In the case of SAN–mRGO2, the electrical conductivity significantly increased from 9.3 × 10^–9^ to 1.2 × 10^–2^ S/m by increasing the mRGO2 concentration from 1.8 wt % to 4.0 wt %. The electrical conductivity was 0.7 S/m at 7.0 wt % of mRGO1. When mGRO1 and mRGO2 were used, the maximum electrical conductivity and a percolation threshold of the obtained nanocomposites were similar values. These results suggested that the degree of reduction of mGO was more important than the reduction methods. To compare with mGRO2, SAN/rGO nanocomposite was prepared through melting mixing of SAN and rGO, the rGO was reduced by solvothermal method. The electrical conductivity of SAN/rGO nanocomposite was 4.5 × 10^–3^ S/m, which the concentration of 3.0 wt % rGO. The electrical conductivity of SAN/rGO was much low than that of SAN–mRGO2 at the same amount of nanosheets. The restacking of layers occurred when the unmodified GO was reduced, which did not effectively disperse in the polymer matrix. (in Appendix A and Figure 7)

The conductivity of SAN–mRGO1 and SAN–mRGO2 starts to become saturated at approximately 7.0 wt % with the value of 0.7 S/m, which indicates that the electrical conductivity of the composites increased significantly once the conductive networks of rGO were formed above a certain critical concentration of rGO in the matrix [44]. Although the electrical properties of mRGOs are not comparable to those of pristine graphene owing to structural defects, there are still many advantages of mRGO for various important applications.

Figure 7 shows SEM images of the fracture surfaces of SAN/rGO and SAN–mRGO2 composites. When the rGO content was 3.1 wt %, the corresponding fracture was uneven however, the rGO sheets were obvious aggregation. In SAN–mRGO2, the mRGO2 sheets were randomly distributed within the SAN matrix even if high content mRGO2 was incorporated. SAN-mGRO2 was observed with only a few layers, but SAN/rGO composite was shown the aggregation of rGO sheets. In case of rGO, restacking occurred during chemical reduction process, however, mRGO2 is thought to have been effectively dispersed without agglomeration occurring due to surface modification of MPS.

## 4. Conclusions

The polymerizable reduced graphene oxide, MPS modified reduced graphene oxides (mRGO) were obtained via the chemical and solvothermal reduction of GO. The structure of mRGOs was confirmed using FT-IR, Raman spectroscopy and XRD. The electrical conductivity of the mRGOs was approximately 1.0 × 10^4^ S/m, which corresponds to half of the theoretical electrical conductivity of graphene (1.0 × 10^7^ S/m).

The grafting of rGO with styrene–acrylonitrile copolymers was prepared via in situ radical polymerization in the presence of mRGOs. The thermal stability and electrical properties of the obtained rGO-grafted copolymers (SAN–mRGOs) were investigated. The degradation temperatures of grafted nanocomposites (*T*_d5%_) are higher than that of neat copolymer. The *T*_d5%_ of nanocomposites increased by more than 20 °C with introduction of mRGOs. A distinct increase in thermal stability was due to mRGO being well-dispersed in the copolymer matrix. The electrical conductivity of SAN–mRGO significantly increased from 9.3 × 10^−9^ to 0.7 S/m by increasing the mRGO2 concentration from 1.8 wt % to 7.0 wt %. The improved electrical and thermal properties are due to good dispersion and interfacial adhesion between graphene and the polymer matrix. When mGRO1 and mRGO2 were used, the thermal degradation temperature and the maximum electrical conductivity of the obtained nanocomposites were similar values. These results suggested that the degree of reduction of mGO was more important than the reduction methods. The radical polymerization process using mRGOs is a promising tool for the preparation of well-dispersed mRGOs within the polymer matrix. It is anticipated that this work will achieve a convenient route to achieving enhanced interfacial interaction. A distinctive and innovative strategy for the covalent functionalization of graphene derivatives has therefore been explored, which may widen the application of 2D nanosheet-based nanocomposites.

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
