# Peer review of "Electrically Conductive Nanocomposites Composed of Styrene–Acrylonitrile Copolymer and rGO via Free-Radical Polymerization"

_polymers, 2020, doi:10.3390/polym12061221_

Round 1
Reviewer 1 Report
In this manuscript, the polymerizable reduced graphene oxide (rGO) was prepare with vinyl substituent alkoxysilane compound to polymerize vinyl monomers and prevent plane restacking, using chemical and solvothermal reduction methods. And then introduced to the electrically conductive styrene-acrylonitrile copolymer via in situ free radical polymerization. Obtained a very high electrical conductivity of 0.7 S/m for the polymer composites. However, there are some problems with the data shown in the manuscript that need to be revised, as detailed below:
- In which specific way do the mRGO improve the electrical conductivity of the styrene-acrylonitrile copolymer contrasted to common rGO? the morphology change? This has to be elaborated on in the main body of the paper. At present this appears to be just a hypothesis!
- How to testify the alkoxysilane substituted to the rGO successfully? How to identify the mRGO is polymerized with styrene-acrylonitrile, rather than blend into styrene-acrylonitrile copolymer? Data describing of the process should be provided
- What the difference between mRGO1 and SAN-mRGO2? Which is the optimal in SAN-mRGO? Why discuss the various contents w% of mRGO2? How about the mRGO1?
- English needs to be further polished.
Author Response
Thanks for your detailed review of this manuscript. According the reviewer’s comments, I responded in turn with a point-by-point.
Comment 1.
SAN/rGO nanocomposite was prepared using common rGO by melt mixing method. The electrical conductivity and morphology were described in text and SEM image shown in Figure 7. The morphology of rGO shown in Figure S3.
This is explained in addition as below. (Revised manuscript, Line 177-282 and Line 293-299)
To compare with mGRO2, SAN/rGO nanocomposite was prepared through melting mixing of SAN and rGO, the rGO was reduced by solvothermal method. The electrical conductivity of SAN/rGO nanocomposite was 4.5 × 10-3 S/m, which the concentration of 3.0 wt% rGO. The electrical conductivity of SAN/rGO was much low than that of SAN-mRGO2 at the same amount of nanosheets. The restacking of layers occurred when the unmodified GO was reduced, which did not effectively disperse in the polymer matrix. (in Figure S3 and Figure 7)
Figure 7 showed SEM images of the fracture surfaces of SAN/rGO and SAN-mRGO2 composites. When the rGO content was 3.0 wt%, the corresponding fracture was uneven however, the rGO sheets were obvious aggregation. In SAN-mRGO2, the mRGO2 sheets were randomly distributed within the SAN matrix even if high content mRGO2 was incorporated. SAN-mGRO2 was observed with only a few layers, but SAN/rGO composite was shown the aggregation of rGO sheets. In case of rGO, restacking occurred during chemical reduction process, however, mRGO2 is thought to have been effectively dispersed without agglomeration occurring due to surface modification of MPS.
Comment 2.
- The alkoxysilane substituted rGO (mGRO2) was confirmed FT-IR. The spectra of mRGO1 and mRGO2 were the same (Figure 1). mGRO1 is what reacts to the MPS on GO before the reduction, and mGRO2 is what reacts to the MPS after the reduction of mGO.
This is explained in addition as below. (Revised manuscript, Line 137-144)
For MPS modified materials, methylene C-H stretching peak appeared at 2920 cm-1 and 2850 cm-1, methyl C-H deformation peak at 1465 cm-1 and Si-O-C stretching peak at 1130 cm-1, also confirming that MPS was successfully introduced on the surface of GO. After the modification of GO with MPS and its reduction, the -OH stretching at 3300 cm-1 disappeared and the characteristic bands at 1300 cm-1 (C-Si stretching) and 1030 cm-1 (Si-O-C stretching) of the mGO appeared [28].
- The grafting efficiency was calculated by Soxhlet extraction with MEK.
This is explained in addition as below. (Revised manuscript, Line 218-224)
In order to measure the grafting efficiency, the nanocomposites obtained by free-radical polymerization was extracted as methyl ethyl ketone (MEK) using a Soxhlet apparatus [35]. Most of the copolymer that was grafted on mGO and mRGO was extracted only 5~6 wt%, while more than 98 wt% of copolymers without mGO and mRGO was extracted. Although there is a limit to calculating grafting efficiency with simple extraction, the obtained grafted copolymer was used to measure thermal stability and electrical conductivity because the extraction amount was only about 5 wt%.
- SAN/rGO nanocomposite was prepared to compare melt blend and in situ polymerization. The results are explained in addition as below. (Revised manuscript, Line 277-282)
To compare with mGRO2, SAN/rGO nanocomposite was prepared through melting mixing of SAN and rGO, the rGO was reduced by solvothermal method. The electrical conductivity of SAN/rGO nanocomposite was 4.5 × 10-3 S/m, which the concentration of 3.0 wt% rGO. The electrical conductivity of SAN/rGO was much low than that of SAN-mRGO2 at the same amount of nanosheets. The restacking of layers occurred when the unmodified GO was reduced, which did not effectively disperse in the polymer matrix. (in Figure S3 and Figure 7)
Comment 3.
- The mRGO1 was prepared by chemical reduction method with hydrazine and mGRO2 was reduced by solvothermal method. The thermal stability and electrical conductivity of SAN-mRGO1 and SAN-mRGO2 were similar at same content of filler in SAN. The results of this study were hard to determine which is optimal, because mGRO1 and mRGO2 have been sufficiently reduced.
The experimental results of mRGO1 with various contents were displayed in Table 2 and Figure 6. As explained above, we could not confirm the difference properties between the results of mRGO1 and mRGO2. Although the reduction method was different, mRGO1 showed a characteristic similar to mRGO2.
Reviewer 2 Report
The paper describes the preparation of reduced GO functionalized with vinyl substituent alkoxysilane which can be conveniently polymerized with styrene and acrylonitrile to provide styrene-acrylonitrile conductive copolymers. The strategy of the work is well-focused and the preparation of the materials well-described. The results are interesting and clearly show the better performance obtained by co-polymerizing the monomers with the functionalized GO with respect to the materials obtained with the un-fuctionalized GO. In my opinion the paper deserves publication after addressing the following minor issues:
1) in the scheme the structure of graphene is wrong
2) the quality of FT-IR spectra reported in figure 2 is very poor and the signals assessing the functionalization as reported in the text are not visible.
At the same time Raman and XRD suggest a disordered structure after functionalization with increased defects and signals at 22.81° and 23.69° in the XRD are not clearly attributed.
Can the authors improve the characterization of mGO and mRGOs to better prove/support the hypothesis about the structure of their modified-GO products?
3) the authors explain the weak improvement in thermal stability of the composites with RGO on the basis of a better distribution of functionalized-GO with respect to material obtained with GO. Could they prove that hypothesis with microscopy?
Author Response
Thanks for your detailed review of this manuscript. According the reviewer’s comments, I responded in turn with a point-by-point.
Comment 1.
The structure of graphene is corrected and shown in Scheme 1.
Comment 2.
- The spectra of GO, mGO, mRGO1 and mRGO2 were measured again with FFT-IR and shown in Figure 1. The assessment of signal was explained in addition as below. (Revised manuscript, Line 137-144)
As shown in Figure 1, FT-IR spectroscopy was used to confirm the structures of GO, mGO and mRGOs and the formation of covalent bonds between the components. The GO exhibited peaks at 1040 cm-1, 1610 cm-1, 1720 cm-1 and 3300 cm-1 corresponding to the C-O, C=C, C=O and -OH groups, respectively, confirming that it contains a variety of oxygen-containing functional groups. For MPS modified materials, methylene C-H stretching peak appeared at 2920 cm-1 and 2850 cm-1, methyl C-H deformation peak at 1465 cm-1 and Si-O-C stretching peak at 1130 cm-1, also confirming that MPS was successfully introduced on the surface of GO. After the modification of GO with MPS and its reduction, the -OH stretching at 3300 cm-1 disappeared and the characteristic bands at 1300 cm-1 (C-Si stretching) and 1030 cm-1 (Si-O-C stretching) of the mGO appeared [28].
- The results of Raman and XRD are described in detail and added to the main text and the supplement information, as below. (Revised manuscript, Line 147-158)
Raman spectroscopy and XRD were performed to determine the modification and reduction of GO and the results are shown in Figure S1 and S2, respectively. The D/G band intensity ratio (ID/IG) of mRGO1 and mRGO2 were 1.37 and 1.13, respectively, which were higher than that of GO (0.98). It indicated that the number of aromatic domains increased after the reduction step (in Figure S1) [29]. After the modification of GO with MPS, the diffraction angle in XRD patterns (in Figure S2) decreased from 11.3° to 10.4°, corresponding to an increase in the interlayer spacing from 0.78 to 0.85 nm. However, these peaks disappeared after the reduction of mGO. These phenomena indicate that MPS is bonded on the surface of GO successfully and is capable of enlarging the interlayer distance between GO sheets. For mRGO1 and mRGO2, a new broad peak can be observed at 22~24°, which are close to that of pristine graphene nanosheet (25°) [30]. The results of Raman spectroscopy and XRD as shown in observation of FT-IR, MPS was successfully introduced on the surface of GO and the chemical and solvothermal method effectively reduced GO.
To observe the morphologies and layers of the obtained GO, rGO, mGO and mRGO, SEM was performed and the images shown in Figure S3.
Comment 3.
- The values of SAN in Figure 5 and Table 2 were mistakenly presented with SAN-mGO values, therefore, the SAN value was modified and inserted Figure and Table. An explanation of thermal stability is given in the text as follows. (Revised manuscript, Line 233-252)
The weight loss curves exhibited a one-step degradation process and shifted slightly to higher temperatures with the introduction of mGO and mRGOs. The degradation temperatures of grafted nanocomposites (Td5%) are higher than that of neat copolymer. For 1.8, 3.0 and 4.7 wt% mRGO2 contained SAN-mRGO2 nanocomposites, the Td5% of composites was 23.5, 24.5 and 28.8 °C higher than that of SAN copolymer. In case of SAN-mRGO1, the Td5% of composites increased by more than 20 °C with introduction of mRGO1. A distinct increase in thermal stability was due to mRGO being well-dispersed in the copolymer matrix. The improvement in thermal stability can be attributed to the delays the escape of volatile degradation products and thus slows down the initial degradation step [36]. The char yield of grafted nanocomposites was slightly higher than the amount of mRGO, and the case of SAN-mGRO2 containing 16 wt% of mRGO2 was 25 wt%. The thermal stability of the obtained nanocomposites was almost same values even with mRGO1, which the loading amount of the MPS is twice as much as mGRO2. These results are not due to MPS modification, but to the high heat capacity and thermal conductivity of mRGO nanosheets, which can form the heat-resistant layers that act as mass transfer barriers [37].
The SAN-mRGO2 nanocomposites have excellent thermal stability, which can be attributed to good interfacial interactions between mRGO2 and SAN. From the SEM image of SAN-mRGO2, the well-dispersed status of mRGO2 in copolymer matrix with little aggregation, suggesting good adhesion between the fillers and matrix. (Figure 7) Due to the interfacial interaction between mRGO2 and copolymer, the char was easy to form on the surface of mRGO sheets, which resulted in the high char yield of nanocomposites [38].
- The dispersion of rRO and mRGO in SAN matrix were confirmed by SEM and shown in Figure 7. The results are described in the Text as below. (Revised manuscript, Line 293-299)
Figure 7 showed SEM images of the fracture surfaces of SAN/rGO and SAN-mRGO2 composites. When the rGO content was 3.0 wt%, the corresponding fracture was uneven however, the rGO sheets were obvious aggregation. In SAN-mRGO2, the mRGO2 sheets were randomly distributed within the SAN matrix even if high content mRGO2 was incorporated. SAN-mGRO2 was observed with only a few layers, but SAN/rGO composite was shown the aggregation of rGO sheets. In case of rGO, restacking occurred during chemical reduction process, however, mRGO2 is thought to have been effectively dispersed without agglomeration occurring due to surface modification of MPS.
Reviewer 3 Report
In the manuscript “Electrically Conductive Nanocomposites Composed of Styrene-Acrylonitrile Copolymer and rGO via in-situ Free-Radical Polymerization”, the authors prepared conductive nanocomposites composed of SAN and polymerizable rGO via in-situ free-radical polymerization. Raman spectra, XRD, FT-IR, TGA, and four-point probe were examined to characterize the properties of the nanocomposites. I think major revisions are needed and the following issues should be addressed by the authors before the manuscript can be considered further.
- Many reports regarding rGO/polymer composites via free-radical polymerization have been published, what is the novelty of this work? In addition, the paper review in the Introduction is not up-to-date and not well organized.
- In the Abstract, the authors mentioned that the elasticity of rGO network and electrical conductivity of nanocomposites improved. However, there is no data related to the elasticity in the manuscript. This is quite an important property of a polymer as a filler is introduced to it. On the other hand, although the conductivity of SAN was improved from 9.8×10-9 to 0.7 S/m, is it good enough for the EMI application mentioned at the end of the manuscript or other applications? Hsiao et al. (ACS Appl. Mater. Interfaces, 6, 10667, 2014) demonstrated lightweight and flexible rGO/PU composites with a conductivity of 16.8 S/m and EMI-shielding effectiveness of 34dB.
- The authors are strongly suggested to input different colors for all figures to easily distinguish different curves.
- Line 156 to 158 on p. 5, Functionalizing the surface of GO with MPS and subsequent reduction processes caused the weight loss of approximately 12 wt% for mGO and approximately 5 wt% for mRGOs over 150~300 °C, mainly attributed to the removal of oxygen-containing functional groups. I think that 12 wt%, 12 wt%, and 5 wt% are for mGO, mRGO1, and mRGO2 respectively.
- Line 159 to 161 on p. 5, The chemically and solvothermally 159 reduced GO exhibited high thermal stability compared to GO, which confirmed that most of the 160 defects in the sp2 plane of GO were restored to the π-conjugated structure. The authors are strongly suggested to provided addition data or references to support this viewpoint of sp2 C-C restoration.
- What is the scale of Y-axis in Fig. 4? Log scale or linear scale?
- The authors are also strongly suggested to provide the morphologies of mGO, rGO, mRGO1, and mRGO2.
Author Response
Thanks for your detailed review of this manuscript. According the reviewer’s comments, I responded in turn with a point-by-point.
Comment 1.
The recent rGO/polymer composites via radical polymerization literature was investigated and written in Introduction Part, and the unique purpose of this paper was also described as below.
(Corrected in Revised manuscript, Line 28-5-28, 37-41 and 45-59)
Graphene oxide (GO) is one of the most promising fillers for nanocomposites, due to its unique electrical, mechanical, optical and thermal properties [1-5] GO sheets were properly investigated due to good dispersity and possible post-functionalization, which are easily reduced, therefore, provide tunability of the electrical conductivity [6,7].
Reduction of GO offers potential to produce few-layered graphene sheets, i.e. reduced GO (rGO), in high amounts. It also offers the possibility for processing of advanced graphene-based materials, consisting of various oxygen containing functional groups and for any application concerning conductive polymer composites [8-10]. There are several reduction processes used to prepare rGO, including chemical reduction, solvothermal reduction, electrochemical reduction, thermal reduction and the use of surfactants or stabilizers. Among these, chemical reduction (especially with hydrazine) and solvothermal reduction are among the most effective and convenient approaches for preparing processable colloidal suspensions required for a broad field of applications [11-13].
rGO has high thermal and electrical conductivity, as rapid transfer is possible between the effectively contacted rGO lamellae, the conductivity of nanocomposites can be significantly improved [14,15]. Therefore, the relative content of rGO within nanocomposites must be increased to form a continuous phase. Unfortunately, aggregation of rGO can easily occur with increasing rGO content. In earlier studies, solution blending, melt mixing, and in-situ polymerization have been used to produce these polymer/rGO composites [16-19]. In-situ polymerization is commonly performed by mixing the filler in the presence of monomers, followed by subsequent polymerization. Upon this method, it is possible to obtain uniformly dispersed polymer/rGO nanocomposites.
In recent years, the grafting of GO sheets with polymer was studied for well-dispersed GO in polymer matrix. Yao et al. [20] grafted polyacrylonitrile using free radical polymerization on gamma irradiation. Yang et al. [21] prepared polystyrene/rGO nanocomposites with radical polymerization, the obtained nanocomposites were enhanced thermal stability on a uniform dispersion of rGO with polymer matrix. Voylov et al. [22] reported that poly(sodium 4-vinylbenzenesulfonate) obtained via RAFT had a controlled molecular weight with a very narrow polydispersity. Sánchez et al. [23] prepared rGO grafted with relatively short chains of poly(n-butyl methacrylate) for electrical field grading materials on high-voltage direct-current applications.
Such in situ polymerization and grafting polymerization methods provides the good dispersion and compatibility of rGO in polymer matrix, and thus excellent thermal, mechanical and electrical properties of the nanocomposites. Researches of polymeric nanocomposite using various monomers have been reported, however few studies gave been done on GO-grafted radical copolymerization of styrene and acrylonitrile. Styrene-acrylonitrile copolymer (SAN) is especially tough and resistant to chemicals, it used a broad range of industries including the food, construction and electrical engineering.
Comment 2.
- I wrote about elasticity of the rGO network by mistake, therefore, deleted it.
- Research on EMI shielding of nanocomposites is currently underway. A study on EMI shielding is working on nanocomposites using various nanosheets, shuc as MXene derivatives and modified MoS2, as well as rGO. The results of EMI will be soon submitted to ‘Polymers’. So please understand that this paper will exclude the results of EMI shielding.
Comment 3.
All the Figures were redrawn using various colors, and presented clearly.
Comment 4 and 5.
The MPS content of mGO, mRGO1 and mRGO2 are calculated using raw data and presented in the text as follows. (Corrected in Revised manuscript, Line 178-183)
Functionalizing the surface of GO with MPS and subsequent reduction processes caused the weight loss of approximately 12 wt% for mGO, 12 wt% for mRGO1 and 5 wt% for mRGO2 over 150~300 °C, mainly attributed to the removal of oxygen-containing functional groups. Further, mRGOs also exhibited higher char yields, 70 wt% for mRGO1 and 80 wt% for mRGO2. The solvothermally reduced GO (mRGO2) showed similar characteristic but with lower amount of weight loss, compared to that of mRGO1. This could be explained by a smaller amount of oxygen functional groups in the surface.
Comment 6.
The electrical conductivities of mGO, mRGO1 and mRGO2 are drawn in log scale and shown in Figure 4.
Comment 7.
- The morphologies of GO, rGO, mGO and mRGO were confirmed by SEM and shown in Figure S3. The results are described in the Supplement information as below. (Revised manuscript, Line 281-282 and Supplement information)
SEM samples of GO, rGO, mGO and mRGO2 were prepared by freezed dried. Figure S3 shows SEM images of exfoliated GO, mGO and mRGO2, which exhibits the typical wrinkle morphology. Meanwhile, the surfaces of mGO and mRGO2 were observed with more wrinkles than that of GO. The rGO was free of wrinkles on the surface and had multiple layers. This is because the layer was restacked during the solvothermal reduction process.
Round 2
Reviewer 1 Report
There has been revised seriously according to the suggestions, I recomend its publication
Author Response
Thank you very much for your detailed, kind review.
It was the most objective and detailed review comment ever experienced. Thanks again for your detailed review.